# Profiling Tight Junction Protein Expression in Brain Vascular Malformations

**DOI:** 10.3390/ijms26104558

**Published:** 2025-05-09

**Authors:** Leire Pedrosa, Alejandra Mosteiro, Luis Reyes, Sergio Amaro, Sebastián Menéndez-Girón, Mateo Cortés Rivera, Carlos J. Domínguez, Anna M. Planas, Ramon Torné, Ana Rodríguez-Hernández

**Affiliations:** 1August Pi i Sunyer Biomedical Research Institute (IDIBAPS), 08036 Barcelona, Spain; lepedrosa@recerca.clinic.cat (L.P.); samaro@clinic.cat (S.A.); anna.planas@iibb.csic.es (A.M.P.); 2Department of Medicine, Faculty of Medicine and Health Sciences, University of Barcelona, 08036 Barcelona, Spain; 3Department of Neurosurgery, Institute of Neuroscience, Hospital Clinic of Barcelona, 08036 Barcelona, Spain; mosteiro@clinic.cat (A.M.); alberto_1704@yahoo.com (L.R.); 4Comprehensive Stroke Center, Institute of Neuroscience, Hospital Clinic of Barcelona, 08036 Barcelona, Spain; 5Department of Neurosurgery, Germans Trias i Pujol University Hospital, Carretera de Canyet, s/n, 08916 Badalona, Spain; s.menendez.94@gmail.com (S.M.-G.); mateocortesr@outlook.es (M.C.R.); cjdominguezalonso@hotmail.com (C.J.D.); 6Department of Neuroscience and Experimental Therapeutics, Institute for Biomedical Research of Barcelona (IIBB), Spanish Research Council (CSIC), 08036 Barcelona, Spain

**Keywords:** blood–brain barrier, brain arteriovenous malformation, tight junctions, vascular permeability

## Abstract

Recent studies suggest that blood–brain barrier (BBB) disruption plays a key role in the clinical course and bleeding risk of brain arteriovenous malformations (bAVMs). The tight junctions (TJs) are complex endothelial transmembrane proteins with a significant physical contribution to BBB disruption. In this study, we hypothesized that bAVMs display a different TJ pattern than other vascular malformations and normal brain tissue. We studied the expression of claudin-5 and occludin as essential factors for functional TJs. Human specimens of surgically resected cavernomas (CCMs) (*n* = 9), bAVMs (*n* = 17), and perilesional brain parenchyma (6 from CCMs and 16 from bAVM patients) were analyzed via immunofluorescence staining, transmission electron microscopy (TEM), and Western blot tests. Compared to perilesional parenchyma, bAVMs showed a significant decrease in TJ protein expression, and these alterations were more apparent in ruptured bAVMs than in unruptured bAVMs or CCMs. TEM images provided evidence of disrupted connectivity between endothelial cells of bAVMs. This is the first clinical investigation that studies the expression of TJs in human bAVMs and their surrounding parenchyma. Despite the limitations of the sample size, we found significant differences in the expression and composition of TJs in bAVMs when compared to surrounding parenchyma and other vascular lesions such as CCMs. These results add further evidence to the role of BBB disruption in the clinical course of bAVM. A deeper understanding of these mechanisms may lead to the development of new therapeutic targets and management strategies for bAVMs.

## 1. Introduction

Brain arteriovenous malformations (bAVMs) are uncommon abnormalities found in the brain tissue which consist of a tangled network of vessels creating a direct connection between arteries and veins, called the nidus [1,2]. This anomalous connection creates a low-resistance circuit that might increase blood flow and pressure, causing stress on the vessel walls and creating an inflammatory environment [3,4]. Over time, this inflammatory environment may evolve and may trigger significant changes in the nidus and the surrounding—or perinidal—brain tissue, thus making bAVMs dynamic lesions with fairly unique features [5].

Up to 60% of bAVMs may present with intracranial bleeding, a serious and feared complication which, moreover, often occurs in young adults [1]. Recent works have shown that elevations of vascular endothelial growth factor (VEGF) related to the aforementioned changes in the vascular wall may contribute to bAVM rupture throughout blood–brain barrier (BBB) disruptions [6,7,8].

The blood–brain barrier (BBB) is a critical component of the neurovascular unit (NVU) that ensures a tightly regulated environment for the central nervous system (CNS). It achieves this by selectively permitting the passage of essential nutrients and molecules while preventing the entry of potentially harmful substances [9,10]. The BBB is primarily formed by endothelial cells, which create a monolayer of tightly joined cells lining the brain capillaries. These endothelial cells are interconnected by continuous tight junctions (TJs), composed of proteins such as occludin and claudins, which play a pivotal role in maintaining the barrier’s integrity [11,12,13,14].

Disruption of the BBB has been implicated in various neurological disorders and diseases. In recent years, there has been a growing appreciation for the potential link between BBB disruption and stroke. While such a link with BBB disruption seems to occur both in hemorrhagic and ischemic stroke, the relationship between cerebral hemorrhage and damage of brain endothelial junctions has been studied much less [15,16,17]. Therefore, understanding the mechanisms underlying BBB disruption in bAVMs seems an intriguing line of research. Further knowledge of the role of TJs in the clinical setting could provide new valuable insights into the development and natural history of bAVMs and could lead to the discovery of potential therapeutic targets.

Considering that bleeding commonly occurs as a complication of bAVMs, we hypothesized that bAVMs may have a unique pattern of TJs when compared to normal brain tissue and to other vascular malformations, like cerebral cavernous malformations (CCMs). For this purpose, we aimed to evaluate the presence of hallmarks of BBB disruption through the investigation of TJ expression in samples from patients with bAVMs and CCMs.

## 2. Results

### 2.1. Demographic Features of the bAVM and CCM Cohort

Viable samples were obtained from 17 sporadic bAVMs and 9 sporadic CCMs. Table 1 and Appendix A summarize the demographic, clinical, and angiographic variables of the bAVM patients. The most frequent form of bAVM onset was intracranial bleeding (6/17; 35%), followed by seizures (5/17; 29%). All patients harbored low- (SM 1–2) (65%) or medium-grade (SM 3) (35%) lesions. Most of the bAVMs were located superficially in the supratentorial compartment (71%), particularly in unruptured cases (82%). Only three patients presented intranidal aneurysms (3/17; 18%). There were no statistical differences in vein diameter or tortuosity, nidus diameter, or ARI score results between ruptured and unruptured bAVMs.

The demographic, clinical, radiological, and treatment-related features of the sample used as a control for BBB disturbances (CCMs) are summarized in Table 2 and Appendix A. Most patients were female (7/9; 78%) and presented with seizures (44%) or with focal deficits (44%). Most CCMs were in the supratentorial superficial compartment (33%). We observed statistical differences in the mean mRS at hospital discharge between ruptured and unruptured CCMs (1.57 vs. 0.00, *p*-value = 0.0171).

### 2.2. Decreased Tight Junctions in bAVMs Compared with CCMs

Immunofluorescence analyses revealed a reduced expression of both claudin-5 and occludin in the blood vessels of bAVMs compared to the corresponding perilesional zone (*p*-value < 0.05), while these differences were not observed in the CCM samples (Figure 1 and Figure 2 and Appendix A). Moreover, occludin expression was lower in bAVM lesion samples compared to CCM lesion specimens (14.29 ± 14.54 vs. 30.08 ± 17.84, *p*-value = 0.0177), whereas no differences were found in claudin expression between both lesions (mean 12.15 ± 8.96 vs. 21.93 ± 16.26 mean, *p*-value = 0.058).

In the WB analysis, no differences between TJ expressions were observed when comparing the lesion samples of bAVMs with the lesion samples of CCMs (Figure 3 and Appendix A). Comparing the perilesional samples of bAVMs with those of CCMs, neither the immunofluorescence of occludin and claudin-5 nor the WB analysis of TJs showed statistically significant differences. Interestingly, according to the WB analyses, the 25 kDa isoform of claudin-5 was only expressed in the perilesional zone in both the bAVMs and CCMs (*p*-value = 0.012) (Appendix A).

### 2.3. Endothelial Cell Morphology and Connections Are Disturbed in bAVMs

TEM images provided evidence of disrupted connectivity between the endothelial cells of bAVMs, indicating the loss of tight junctions (Figure 4), which aligned with the findings from the immunofluorescence and WB assays. Endothelial cells exhibited membrane alterations, such as the presence of endosomes and empty vacuoles in the cytosol. Additionally, the conformation of collagen and fibronectin fibers was suggestive of stress and cell deterioration.

### 2.4. Differences in the Perinidal Tight Junctions of Ruptured vs. Unruptured bAVMs

The expression of TJ proteins differed significantly between samples obtained from the perinidal tissue of ruptured and unruptured bAVMs. A decreased expression of occludin and claudin-5 was found in the parenchyma surrounding the ruptured bAVMs in comparison with the perilesional zone of unruptured bAVMs. Similarly, ruptured bAVM lesions showed a non-significant decrease in TJ expression compared to unruptured bAVMs, as assessed by immunofluorescence testing (Figure 5). The WB analysis also showed a decrease in the expression of occludin and claudin-5 in the perilesional zone of ruptured AVMs, although this difference was also not statistically significant (Figure 6). Finally, the expression of TJs did not differ between ruptured and unruptured bAVM lesion samples.

## 3. Discussion

This is the first clinical investigation that studies the expression of TJs in human bAVMs and their surrounding parenchyma. Despite the limitations of the sample size, we found significant differences in the expression and composition of TJs in bAVMs when compared to surrounding parenchyma and other vascular lesions such as CCMs. These results add further evidence to the role of BBB disruption in the clinical course of bAVM.

### 3.1. BBB Disruption in bAVMs

Tight junctions play a crucial role in maintaining the integrity of the BBB [18]. The disruption of TJs can lead to dysfunction of the BBB, allowing harmful substances to enter the brain [13,19,20,21]. In experimental models, BBB dysfunction has been implicated in the pathogenesis of various cerebral vascular pathologies [22,23,24]. In line with these previous findings, we found an irregular and interrupted arrangement of endothelial cell-contact proteins in bAVMs compared to the control tissue. The expression of occludin and claudin-5 in endothelial cells from bAVM specimens was decreased both in the immunofluorescence and WB assays. Taken together, these observations suggest that bAVMs may harbor significant alterations in their TJs, which might result in BBB disruption. Endothelial dysregulation leading to neovascularization and angiogenesis is commonly seen in bAVMs and could promote an imbalance in the stability of the lesion, and thus lead to edema or bleeding [25]. One of the major underlying molecular mechanisms of endothelial dysregulation and BBB disruption is the activation of matrix metalloproteinases (MMPs), which may lead to degradation of the vascular basal lamina, cerebral edema, and hemorrhage [26,27]. Similar to our clinical findings, Ma et al. observed a decreased expression of claudin-5 in the endothelial cells of bAVMs in mice. Claudin-5 seems to be a particularly crucial claudin for BBB occlusion since claudin-5-knockout (KO) mice die hours after birth from brain edema due to BBB disruption [28]. Similarly, another study with claudin-KO mice models also suggests that claudin-5 is uniquely important for barrier occlusion [25]. Overall, these data suggest that bAVMs present alterations in TJs and that this might be involved in BBB disruption. Further, such BBB disruption might provide an explanation for the occurrence of bleeding in these pathologies.

Claudin-5 and occludin are vital tight junction proteins within endothelial cells, primarily responsible for maintaining vascular integrity and regulating the permeability of blood vessels [12]. While their well-known function involves forming tight junctions that control paracellular transport, emerging evidence suggests that these proteins may play additional roles in brain endothelial cells [29]. These non-canonical functions offer important insights into endothelial cell dynamics and how they might contribute to AVM pathology.

One notable role of claudin-5 and occludin is their involvement in signal transduction, where they may mediate cellular responses to mechanical stress and inflammation [30], which are frequently present in AVM lesions. Additionally, claudin-5 and occludin may influence endothelial cell plasticity, aiding either in maintaining cellular identity or facilitating the dedifferentiation seen in AVM lesions [31]. This cellular flexibility could further impact the development and progression of these malformations.

Beyond these functions, claudin-5 and occludin are also implicated in cytoskeletal interactions, contributing to endothelial cell stability and shape [32]. Therefore, disruptions in these interactions in AVM lesions may compromise vascular integrity and exacerbate the pathological progression of the lesions. Moreover, these proteins could play roles in abnormal angiogenesis and modulate endothelial responses in hypoxic or inflammatory environments, thereby affecting permeability and immune cell recruitment. Understanding the dysfunction and non-canonical roles of claudin-5 and occludin provides valuable perspectives for exploring therapeutic strategies targeting endothelial dysfunction in AVM pathology.

### 3.2. BBB Disruption Can Be Analyzed via TEM

The first paper that studied the ultrastructural features of vascular malformations using electron microscopy was published in 2000 by John H. Wong et al. [33]. In their study, three CCMs and three AVMs microsurgically harvested from patients were analyzed via TEM. Normal control cerebral tissue was obtained from two patients undergoing surgery for epilepsy. Specific attention was directed at components of the vascular wall, endothelial cell (EC) morphology, intercellular tight junctions, and the subendothelial layer. They observed several differences between bAVM and CCM specimens. In embolized bAVM vessels, ECs were disrupted and the nidal vessel walls showed disorganized collagen bundles, but the tight junctions were preserved. On the other hand, in the CCM specimens, ECs lined attenuated cavern walls that were composed of an amorphous material lacking organized collagen. These findings suggested that the BBB was disrupted in both pathologies, although there were different patrons of alteration between pathologies. Tu et al. also observed through TEM analysis that the size and thickness of blood vessels vary in bAVM specimens compared to healthy blood vessels [34]. They observed that the elastic lamina was split, the endothelial layer was thickened, and increased collagen and lymphocyte infiltration were present in the vessel walls. In addition, they observed that, in bAVMs, the endothelium was discontinuous and there were no pericytes, and their endothelial cells had fenestrated processes, filopodia, lysosomes, and cytoplasmic vesicles. Herein, we expanded these previous descriptions to include a larger number of bAVMs and CCMs. Similar to previous work in the literature, our results reveal a compromised BBB in the abnormal vessels of bAVMs. Our TEM images provide evidence of disrupted connectivity between the endothelial cells of bAVMs, indicating the loss of tight junctions. Moreover, our TEM findings align with those of our immunofluorescence and WB analyses, providing further proof of the importance of BBB disruption in these vascular lesions.

### 3.3. Hemorrhage and BBB Disruption

Hemorrhage is probably the most feared clinical presentation of bAVMs since it leads to up to 30% rates of severe morbidity and mortality in young adults. Several studies have shown that proteins involved in maintaining the BBB are significantly decreased after hemorrhage, suggesting a decrease in the expression of TJs in the surrounding parenchyma after a hemorrhagic event in conditions such as bAVM or CCM.

According to our data, the perinidal parenchyma around a ruptured bAVM expresses fewer TJs than the perinidal parenchyma around an unruptured bAVM, which further supports that the BBB might be disrupted after bleeding. Studies with rodent and endothelial cell cultures suggest that elastase generated during cerebral damage associated with bleeding might interact with protease-activated receptors and [Ca_2_^+^]_i_ signalization in brain endothelial cells, which would have contributed to the disruption of the blood–brain barrier that we were able to observe in our clinical samples. Nevertheless, having proof from clinical studies of BBB disruption after bAVM bleeding can aid the development of further research on treatment strategies to prevent secondary brain damage in these patients.

### 3.4. BBB Disruption in CCMs

In our study, CCMs were used as a control group. Comparing the TJs in bAVMs and CCMs, we observed that the endothelial cells of bAVMs expressed fewer TJs than the endothelial cells of CCMs, especially occludin. These findings seem to be in agreement with previous studies in the literature regarding expression patterns of TJ proteins in CCMs. Li Ma et al.’s study also suggested that bAVMs show a higher level of impairment of vascular integrity than CCMs [35]. According to several reports, CCMs may exhibit alterations in their TJ complex assembly, manifesting as a loss of the critical TJ protein–protein interactions that are crucial for a stable junctional complex, despite the fact that the expression profile of the TJs might not be directly affected [36,37,38]. Interestingly, our results suggest the presence of a higher expression of occludin and claudin-5 in the blood vessels of CCMs compared with their perilesional tissue. Burkhardt et al. also observed an overexpression of tight junction proteins in CCM specimens compared to surrounding normal brain tissue. In addition, this overexpression of TJs appeared to be associated with specific clinical patient phenotypes characterized by previous bleeding [39]. This might be explained by the constant micro-bleeding from CCMs, which might impair the BBB in their perilesional tissue and thus lead to the aforementioned findings. Nonetheless, the expression pattern of TJs in CCMs appears to be associated with specific clinical patient phenotypes characterized by the presence or absence of previous bleeding [39]. Therefore, larger studies with more patients and detailed clinical data might shed further light on this topic.

### 3.5. Limitations

A primary limitation of this study is the reliance on Western blotting techniques, which often utilize whole-brain samples rather than isolated microvessels or endothelial cells. This approach may obscure specific expression patterns of junction proteins. Furthermore, the discrepancies between our Western blot and immunofluorescence results indicate that variations in protein expression may go undetected, as immunofluorescence provides more precise visualization in vascular structures. Another significant limitation is the inherent variability among biological samples, which can affect the reproducibility and reliability of our findings. Differences in genetic background, environmental conditions, and individual health status may introduce substantial variability, complicating the detection of true effects and hindering definitive conclusions. Additionally, the limited amount of tissue available for analysis restricted the number of samples processed, reducing the statistical power of the experiments. A smaller sample size increases the risk of sampling bias and may not adequately represent the broader population. Moreover, the necessity of pooling samples may mask individual variations and introduce confounding factors.

## 4. Materials and Methods

### 4.1. Patient Selection

We carried out a histological analysis of brain tissue samples obtained in a prospective study, evaluating the role of BBB dysfunction in the course of surgically treated bAVMs by means of clinical, radiological, and histological testing. Our clinical and radiological findings have been reported in a separate paper [40]. Patients with both ruptured and unruptured bAVMs, eligible for surgery, were sequentially included. Two tertiary neurosurgical-referral centers participated in our study, beginning in January 2020 with recruitment being completed in December 2023. The comparison group consisted of a sequential series of patients harboring CCMs, both ruptured and unruptured, who were considered candidates for microsurgical excision.

The study protocol was approved by the Clinical Ethics Committee of both participating centers and complies with national and European law regarding biomedical investigations with human subjects, along with the 1975 Declaration of Helsinki and subsequent amendments. Patients, or their legal representatives, gave their written consent before inclusion in the study.

Ruptured bAVMs were defined as those diagnosed after acute rupture or with evidence of previous bleeding. Unruptured bAVMs were defined as those discovered incidentally (asymptomatic) or because of associated symptoms (seizures, focal neurological deficits) not related to intracranial bleeding. Inclusion criteria for ruptured bAVM patients were as follows: age ≥ 18 years old and acute intracranial bleeding or evidence of previous intracranial hemorrhage. Inclusion criteria for unruptured bAVM patients were as follows: age ≥ 18 years old and symptoms related to bAVM in the absence of prior intracranial bleeding (unruptured symptomatic) or a bAVM discovered incidentally (unruptured, asymptomatic). Exclusion criteria (for all subgroups) were as follows: unstable clinical condition; premorbid disability; pregnancy; impaired renal function; MRI contraindications.

### 4.2. Clinical, Angiographic, and Neuropsychological Evaluation

Demographics and clinical presentation (seizures, headaches, neurological deficits, or asymptomatic) were recorded before surgery (T0). The Spetzler–Martin grading scale was used to categorize the bAVMs into low grade (SM = 1–2), medium grade (SM = 3), or high grade (SM = 4–5). Lawton–Young’s supplementary score was also noted. The functional and neurological statuses were recalled at baseline (T0) (a premorbid state in case of ruptured bAVMs) and at follow-up within 3, 6, and 12 months after treatment through the modified Rankin scale (mRS).

Several angiographic features of the bAVMs were recorded, including the location (supratentorial superficial/deep, paraventricular, or infratentorial), nidus diameter, intranidal/flux aneurysms, venous drainage pattern, and characteristics of the main draining vein (diameter, length, stenosis or ectasia, number and angulation of the curves). The AVM rupture index (ARI) was noted in each case as a summative punctuation of three parameters (if present): (1) tortuosity (any curve > 180° in the draining vein); (2) single venous drainage; (3) paraventricular or infratentorial location [2].

### 4.3. Brain Samples

During the surgical procedure, the neurosurgeon carefully extracted tissue samples from the patient and identified their specific anatomical regions. This process involved distinguishing the arteriovenous malformation (AVM), characterized by direct connections between arteries and veins without the presence of capillaries, from the perilesional or perinidal tissue. This identification ensured precise categorization of the samples for subsequent analysis.

After surgical resection, the specimens were fixed and cryoprotected. The AVM nidus was identified in all samples and used to recognize the area of interest for immunolabeling and Western blotting studies. The surrounding parenchyma (perilesional zone) was separated from the malformation and then frozen or fixed separately. Brain samples were washed 3 times with sterile PBS 1X and distributed to the tubes for posterior laboratory analyses. Appendix A shows the protocol design for laboratory sample processing.

### 4.4. Immunofluorescence

Brain tissue was washed 3 times with PBS 1X, fixed with PFA at 4%, for 72 h at 4 °C, cryoprotected in 30% sucrose (24 h at 4 °C), and then frozen at −80 °C. Cryostat brain sections (12 μm thick) were fixed with ethanol at 70%, blocked with 3% normal donkey serum, and incubated overnight at 4 °C with either a monoclonal primary antibody against occludin (#71-1500, ThermoFisher, Waltham, MA, USA), diluted 1:125 (2 µg/mL), or claudin-5 conjugated with Alexa Fluor™ 488 (#352588, ThermoFisher, USA), diluted 1:50. Then, sections were incubated for 1:30 h at room temperature with appropriate secondary antibodies (Alexa Fluor 546; Molecular Probes; Life Technologies, S.A., Eugen, OR, USA). Immunoreaction controls were carried out by omission of the primary antibodies. Sections were counterstained with 4′,6-diamidino-2-phenylindole (DAPI) (Invitrogen, Waltham, MA, USA) to visualize the cell nuclei and were observed under a Dragonfly confocal microscope (Andor, Oxford Instruments, Belfast, Northern Ireland, UK).

Image analysis of occludin and claudin-5 staining was performed with Fiji software (ImageJ v1.53). Quantification of fluorescence in the blood vessel (% area) was carried out by determining the area of fluorescence in every blood vessel observed in each sample at 4× magnification. In brief, we generated a z-projection for each fluorescence channel. Segmentation was performed in the channel of DAPI and the corresponding mask was applied to the occludin or claudin-5 channel to measure the percentage of positive expression in blood vessels. The negative controls were used to select the threshold of positive expression for each channel.

### 4.5. Western Blotting

Brain samples were immediately frozen on dry ice and stored at −80 °C after 3 washes with PBS 1X. Tissue was homogenized in radio-immunoassay precipitation (RIPA) buffer and centrifuged for 20 min at 12,000× *g* at 4 °C, and the supernatant was used for protein determination according to the Bradford method. A total of 25 μg of protein was mixed with a loading buffer containing dithiothreitol (DTT) and samples were loaded in 4–15% polyacrylamide gradual gels for electrophoresis (#MB46601, NZYTECH, Lisbon, Portugal). Proteins were then transferred to polyvinylidene fluoride membranes (Immobilon-P, #IPVH00010, Millipore/Sigma, Burlington, MA, USA) and incubated overnight at 4 °C with primary antibodies, followed by horseradish peroxidase-conjugated secondary antibodies for 1 h at RT. Primary antibodies were against occludin (#71-1500, ThermoFisher; 1:250) and claudin-5 (#35-2500, ThermoFisher; 1:500). GAPDH (#ab181602, Abcam, Cambridge, UK; 1:10,000) was used as a loading control. Blots were developed with a chemiluminescent substrate (ECL Amersham, # RPN2235, Buckinghamshire, UK) and visualized with the ChemiDoc MP Imaging System (Bio-Rad Laboratories, Inc., Hercules, CA, USA). Band intensity was quantified by quantity one (Bio-Rad).

### 4.6. TEM

Tissues were cut into small pieces and fixed with 2.5% glutaraldehyde and 2% paraformaldehyde in 0.1 M phosphate buffer. Samples were post-fixated with osmium tetroxide and dehydrated with acetone, embedded in resin, and sectioned using the Leica ultramicrotome UC7 (Leica Microsystems, Wetzlar, Germany). Ultrathin sections (50–70 nm) were stained with 2% uranyl acetate for 10 min and a lead–citrate staining solution for 5 min, and then analyzed with a JEOL JEM-1010 transmission electron microscope fitted with a Gatan Orius SC1000 digital camera (model 832) at the TEM-SEM Electron Microscopy Unit within the Scientific and Technological Centers of the University of Barcelona (CCiTUB).

### 4.7. Statistical Analysis

Statistical analysis was conducted in SPSS v.27 (IBM Corp., New York, NY, USA), GraphPad Prism v.5, and R Studio software v.4.2.0. Data were summarized according to the mean (±standard deviation) and proportions. Two-tailed tests were applied to detect significance between groups, selected according to parametric or non-parametric data, the number of groups, and their distribution. A paired *t*-test was applied comparing samples of malformation and the perilesional zone from the same patient. A *p*-value of <0.05 was established for rejection of the null hypothesis. Histograms were obtained using R Studio.

## 5. Conclusions

Our findings build upon previous experimental studies revealing substantial differences in the expression and composition of tight junctions (TJs) in human bAVM samples when compared to their perinidal tissue and other vascular lesions, such as CCMs. These results contribute critical new knowledge to the understanding of bAVM physiopathology, showcasing the profound impact of TJ alterations on the integrity of the blood–brain barrier (BBB). The discovery that bAVMs express reduced levels of TJs underscores the vulnerability of the BBB in these lesions, providing essential insights that may pave the way for innovative therapeutic strategies to protect BBB function and improve clinical outcomes for patients.

While our study does face several limitations, they do not diminish the importance of these findings. Our reliance on Western blotting techniques, which involve whole-brain samples rather than isolated microvessels or endothelial cells, may have obscured specific patterns of TJ protein expression. Nevertheless, our data demonstrate significant differences that highlight the disrupted BBB in bAVMs. Discrepancies between the Western blotting and immunofluorescence results further emphasize the complexity of TJ protein expression; however, immunofluorescence offers precise visualization that supports our conclusions. Furthermore, biological variability among samples—due to genetic, environmental, and health-related factors—may present challenges, but the consistent reduction in TJ expression across bAVM samples stands out as a key finding. The limitations of smaller sample sizes and the necessity of pooling samples, while acknowledged, do not detract from the overall significance of identifying reduced TJ expression in bAVMs and its implications for BBB disruption.

Importantly, this study sets the stage for future investigations into the molecular dynamics of bAVMs. By employing advanced methodologies, such as isolating specific cell populations or increasing the sample size, subsequent research can further refine and validate our findings. We anticipate that the insights provided by this study will drive novel perspectives in understanding BBB dysfunction in bAVMs, inspiring innovative approaches to therapeutic development. Ultimately, the work presented here provides a solid foundation for translating molecular discoveries into clinical interventions, improving the quality of life and outcomes for bAVM patients.

## Figures and Tables

**Figure 1 ijms-26-04558-f001:**
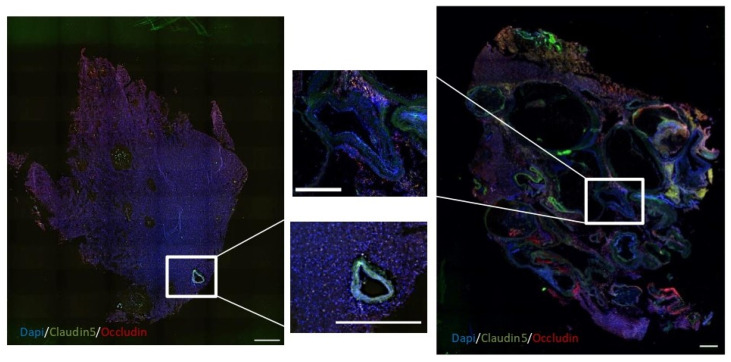
TJ protein expression analyzed with immunofluorescence. Images obtained via fluorescence microscopy (4×) after immunostaining of the TJ protein expression in the perilesional zone of a bAVM (**left**) and the lesion of a bAVM (**right**). A representative blood vessel is amplified from the middle panel from each sample (square). Claudin-5 (green) and occludin (red) were used to mark tight junctions and endothelial cells, respectively. The nucleus was marked with DAPI (blue).

**Figure 2 ijms-26-04558-f002:**
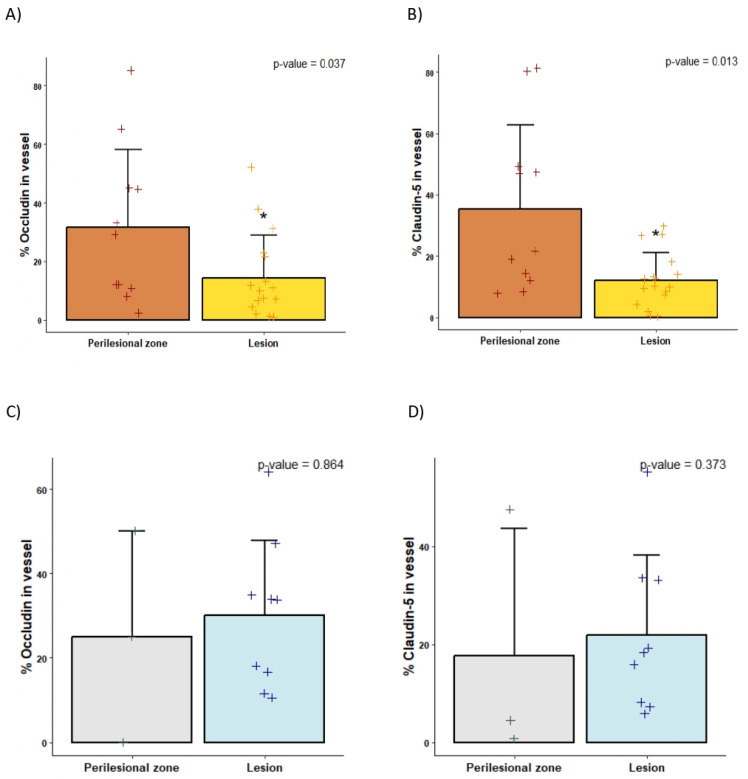
Quantification of TJ protein expression analyzed with immunofluorescence. Histograms of claudin-5 (**right**) and occludin (**left**) protein expression in vessels of the perilesional zone and lesion zone of bAVM (**A**,**B**) and CCM (**C**,**D**) specimens, quantified by ImageJ v1.53. *n* = 14 perilesional zone; *n* = 26 lesion; *n* = 3 CCM. * *p*-value < 0.05. +: values of each sample in each group.

**Figure 3 ijms-26-04558-f003:**
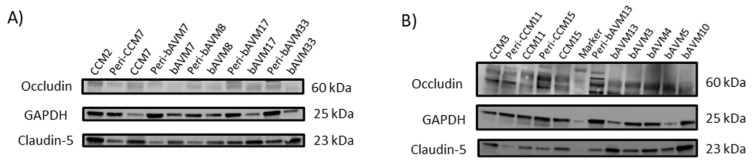
TJ protein expression analyzed via Western blotting. Western blotting results (**A**,**B**). bAVM: brain arteriovenous malformation; CCM: cerebral cavernous malformation; Peri: Perilesional zone.

**Figure 4 ijms-26-04558-f004:**
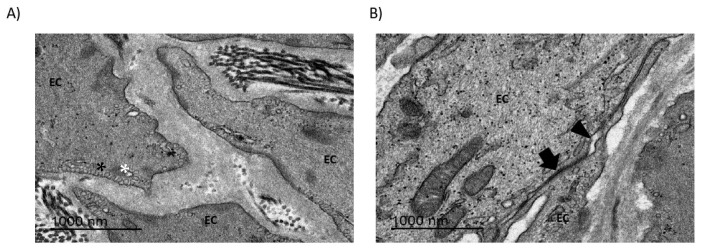
TEM images of bAVMs. Gaps (arrowhead) are observed in the connection between endothelial cells. (**A**) 15,000×. (**B**) 5000×. Black asterisk: endosomes; white asterisk: empty vacuoles; arrow: tight junctions; EC: endothelial cell.

**Figure 5 ijms-26-04558-f005:**
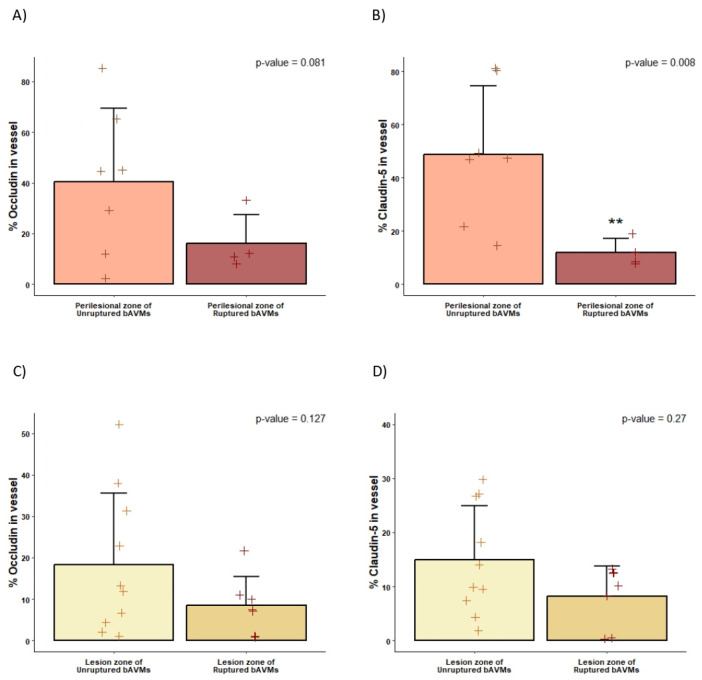
Quantification of claudin-5 and occludin analyzed via immunofluorescence. Histograms of claudin-5 (**right**) and occludin (**left**) protein expression in vessels of unruptured and ruptured samples of the perilesional zone (**A**,**B**) and the lesion zone of bAVM (**C**,**D**) specimens, quantified by ImageJ. **: *p*-value < 0.01. +: values of each sample in each group.

**Figure 6 ijms-26-04558-f006:**
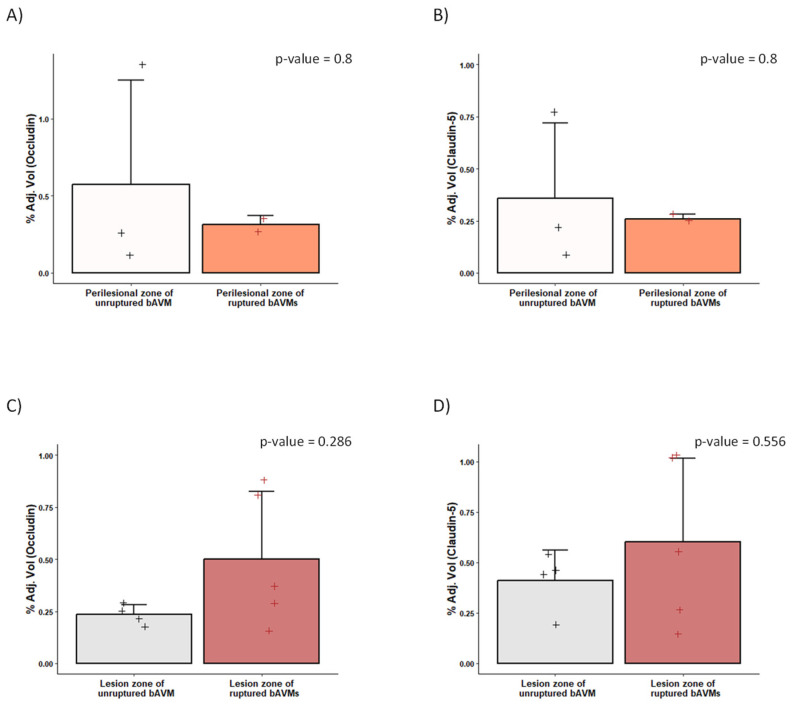
Quantification of claudin-5 and occludin analyzed via Western blotting. Histograms of claudin-5 (**right**) and occludin (**left**) protein expression in unruptured and ruptured samples of the perilesional zone (**A**,**B**) and the lesion zone of bAVM (**C**,**D**) specimens, quantified by Onelab. +: values of each sample in each group.

**Table 1 ijms-26-04558-t001:** Clinical and radiological features of bAVM patients. Data are given globally for all brain arteriovenous malformations (bAVMs) and dichotomized according to the rupture status at diagnosis.

	All bAVMs (*n* = 17)	Ruptured bAVMs (*n* = 6)	Unruptured bAVMs (*n* = 11)	*p*-Value
Age	43.47 (13.29)	41.83 (18.14)	44.36 (10.75)	0.763
Sex, female (%)	9 (52.94%)	2 (33.33%)	7 (63.64%)	0.330
Clinical onset				0.018
Asymptomatic	3 (17.65%)	1 (16.67%)	2 (18.18%)	
Headache	1 (5.88%)	0 (0.00%)	1 (9.09%)	
Seizures	5 (29.41%)	0 (0.00%)	5 (45.45%)	
Focal deficit	2 (11.76%)	0 (0.00%)	2 (18.18%)	
Bleeding	6 (35.29%)	5 (83.33%)	1 (9.09%)	
Location				0.140
Supratentorial superficial	12 (70.59%)	3 (50.00%)	9 (81.82%)	
Supratentorial deep	0 (0.00%)	0 (0.00%)	0 (0.00%)	
Paraventricular	3 (17.65%)	1 (16.67%)	2 (18.18%)	
Infratentorial	2 (11.76%)	2 (33.33%)	0 (0.00%)	
SM, *n* (%)				1.000
1	4 (23.53%)	1 (16.67%)	3 (27.27%)	
2	7 (41.18%)	3 (50.00%)	4 (36.36%)	
3	6 (35.29%)	2 (33.33%)	4 (36.36%)	
Nidus diameter (mm)	26.24 (9.28)	26.45 (2.74)	25.83 (4.28)	0.900
Intranidal aneurysms, *n* (%)	3 (17.65%)	2 (33.33%)	1 (9.09%)	0.510
Venous drainage, *n* (%)				0.322
Only deep	2 (11.76%)	1 (16.67%)	1 (9.09%)	
Not only deep	3 (17.65%)	2 (33.33%)	1 (9.09%)	
Superficial	12 (70.59%)	3 (50.00%)	9 (81.82%)	
Vein diameter (mm)	3.57 (1.45)	3.43 (1.61)	3.64 (1.43)	0.791
N° curves 45–90°	1.29 (1.05)	1.33 (0.82)	1.27 (1.19)	0.903
N° curves 90–180°	1.35 (0.93)	1.17 (0.75)	1.45 (1.03)	0.522
N° curves > 180°	1.23 (1.39)	2.00 (1.67)	0.82 (1.078)	0.160
Venous length (mm)	72.77 (46.22)	72.76 (51.49)	72.77 (45.74)	0.999
ARI				0.756
0–1	9 (52.94%)	3 (50.00%)	6 (54.55%)	
2–3	8 (47.06%)	3 (50.00%)	5 (45.45%)	
Treatment type				0.600
Surgery	6 (35.29%)	3 (50.00%)	3 (27.27%)	
Endovascular	0 (0.00%)	0 (0.00%)	0 (0.00%)	
Combined	11 (64.71%)	3 (50.00%)	8 (72.73%)	

Data are given as the mean (standard deviation) unless otherwise specified.

**Table 2 ijms-26-04558-t002:** Clinical, radiological, and treatment-related variables of the control group (cavernomas, CCMs).

	All CCMs (*n* = 16)	Ruptured CCMs (*n* = 9)	Unruptured CCMs (*n* = 7)	*p*-Value
Age	48.11 (15.66)	50.43 (15.28)	40.00 (19.79)	0.5893
Sex, female (%)	7 (77.78%)	6 (85.71%)	1 (50.00%)	0.416
Clinical onset				0.5556
Asymptomatic	0 (0.00%)	0 (0.00%)	0 (0.00%)	
Headache	1 (11.11%)	1 (14.29%)	0 (0.00%)	
Seizures	4 (44.44%)	2 (28.57%)	2 (100.00%)	
Focal deficit	4 (44.44%)	4 (57.14%)	0 (0.00%)	
Location	0			0.5556
Supratentorial superficial	3 (33.33%)	2 (28.57%)	1 (50.00%)	
Supratentorial deep	1 (11.11%)	0 (0.00%)	1 (50.00%)	
TE	2 (22.22%)	2 (28.57%)	0 (0.00%)	
Cerebellum	2 (22.22%)	2 (28.57%)	0 (0.00%)	
Multiple	1 (11.11%)	1 (14.29%)	0 (0.00%)	
Diameter (mm)	25.67 (6.06)	27.14 (6.07)	20.50 (2.12)	
Multiple, *n* (%)	4 (44.44%)	3 (42.86%)	1 (50.00%)	1
Treatment type, surgery, *n* (%)	9 (100.00%)	7 (100.00%)	2 (100.00%)	

## Data Availability

Data is contained within the article and Appendix A.

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
