# Peer review of "Profiling Tight Junction Protein Expression in Brain Vascular Malformations"

_ijms, 2025, doi:10.3390/ijms26104558_

Round 1

Reviewer 1 Report

Comments and Suggestions for Authors

The study is of interest and worth publication in ijms. A few points should be fixed before acceptance. The conclusion (section 5) should be developed, explain who the authors anticipate the perspectives, and the conclusions merged with section 6. The limitations of the study are an added-value for the manuscript and should be discussed.

Too many abbreviations render the manuscript difficult to follow, starting from the abstract. The reviewer suggest not to use the abbreviation BBBd and prefer BBB disruption throughout the text (the authors did so line 177). 

Reference 1 is not used in the text, starting from ref 2.

lines 52-56: It would be nice to revise this sentence, to site other references and introduce the neuro-vascular unit (NVU). Indeed the central elements of BBB are the endothelial cells that form a monolayer of tightly joined cells covering the brain capillaries. TJs are part of the endothelial cells.

Figure 3: panels C to F are not give more info than in panels A-B and could be removed or add as supplementary figure. This figure could be merged with Figure 4.

Caption of figure 5: a is used for p-value <0,005 but the figure displays a , which is the corresponding p-value ?

Tables should be simplified and the detailed ones added as supplementary tables

Author Response

We sincerely thank the reviewer for their thorough evaluation and valuable feedback on our manuscript. Your insightful comments have greatly contributed to improving the quality and clarity of our work.

Comment 1: The study is of interest and worth publication in ijms. A few points should be fixed before acceptance. The conclusion (section 5) should be developed, explain who the authors anticipate the perspectives, and the conclusions merged with section 6. The limitations of the study are an added-value for the manuscript and should be discussed.

Response 1: Thank you for the reviewer's valuable feedback. In response, we have expanded the conclusions to include a detailed discussion of the study's limitations:

“Our findings build upon previous experimental studies, revealing substantial differences in the expression and composition of tight junctions (TJs) in human bAVM samples when compared to their perinidal tissue and other vascular lesions, such as CCMs. These results contribute critical new knowledge to the understanding of bAVM physiopathology, showcasing the profound impact of TJ alterations on the integrity of the blood-brain barrier (BBB). The discovery that bAVMs express reduced levels of TJs underscores the vulnerability of the BBB in these lesions, providing essential insights that may pave the way for innovative therapeutic strategies to protect BBB function and improve clinical outcomes for patients.

While our study does face several limitations, they do not diminish the importance of these findings. The reliance on Western blotting techniques, which involve whole-brain samples rather than isolated microvessels or endothelial cells, may obscure specific patterns of TJ protein expression. Nevertheless, our data demonstrate significant differences that highlight the disrupted BBB in bAVMs. Discrepancies between Western blot and immunofluorescence results further emphasize the complexity of TJ protein expression, yet immunofluorescence offers precise visualization that supports our conclusions. Furthermore, biological variability among samples may present challenges, but the consistent reduction in TJ expression across bAVM samples stands out as a key finding. The limitations of smaller sample sizes and the necessity of pooling samples, while acknowledged, do not detract from the overall significance of identifying reduced TJ expression in bAVMs and its implications for BBB disruption.

Importantly, this study sets the stage for future investigations into the molecular dynamics of bAVMs. By employing advanced methodologies, such as isolating specific cell populations or increasing sample size, subsequent research can further refine and validate our findings. We anticipate that the insights provided by this study will drive novel perspectives in understanding BBB dysfunction in bAVMs, inspiring innovative approaches to therapeutic development. Ultimately, the work presented here provides a solid foundation for translating molecular discoveries into clinical interventions, improving the quality of life and outcomes for bAVM patients.”

Comment 2: Too many abbreviations render the manuscript difficult to follow, starting from the abstract. The reviewer suggests not to use the abbreviation BBBd and prefer BBB disruption throughout the text (the authors did so line 177). 

Response 2: We totally agree with the reviewer that BBB disruption might clarify the text and make it easier to follow. Therefore, we have changed the abbreviations BBBd to BBB disruption.

Comment 3: Reference 1 is not used in the text, starting from ref 2.

Response 3: We have reviewed and corrected the references. Thank you for the observation.

Comment 4: lines 52-56: It would be nice to revise this sentence, to site other references and introduce the neuro-vascular unit (NVU). Indeed the central elements of BBB are the endothelial cells that form a monolayer of tightly joined cells covering the brain capillaries. TJs are part of the endothelial cells.

Response 4: We agree with the reviewer that we should explain and reference the NVU and the elements of BBB. Therefore, we changed the lines 52-56 to the following text, including new references:

“The blood-brain barrier (BBB) is a critical component of the neurovascular unit (NVU), which ensures a tightly regulated environment for the central nervous system (CNS). It achieves this by selectively permitting the passage of essential nutrients and molecules while preventing the entry of potentially harmful substances10,11. The BBB is primarily formed by endothelial cells, which create a monolayer of tightly joined cells lining the brain capillaries. These endothelial cells are interconnected by continuous tight junctions (TJs), composed of proteins such as occludin and claudins, which play a pivotal role in maintaining the barrier's integrity12,13.”

Comment 5: Figure 3: panels C to F are not give more info than in panels A-B and could be removed or add as supplementary figure. This figure could be merged with Figure 4.

Response 5: We appreciate the observation and we have modified the figures according to the reviewer suggestions.

Comment 6: Caption of figure 5: a is used for p-value <0,005 but the figure displays a, which is the corresponding p-value?

Response 6: Thank you for bringing this to our attention. We have removed the mention of p-value < 0.005 from the figure legend, as it was unnecessary to include. This adjustment has been made since there are no values below 0.005 in the data.

Comment 7: Tables should be simplified and the detailed ones added as supplementary tables

Response 7: We appreciate this thoughtful observation. Accordingly, tables 1 and 2 have been simplified and most detailed outcome data is now provided as supplementary material.

Reviewer 2 Report

Comments and Suggestions for Authors

Pedrosa and coauthors investigated the tight junction profile expression in brain arteriovenous malformation (AVM) and cerebral cavernous malformation ( CCM). The analysis included human specimens of surgically resected CCMs and  AVM lesions and perilesion areas. Their results showed that bAVM had decreased Tj protein, claudin-5, and occludin expression in ruptured bAVM compared to unruptured bAVM or CCMs.           

This is a good study on an important topic.  The manuscript is well prepared, and the experiments were well designed and mapped out.  The data is convincing, and well-presented.

There are several points which are very important to address:

  1. In AVM, the capillary bed, a site of the blood-brain barrier (BBB), is bypassed. However, a specific group of AVMs, the capillary AVM, could affect the capillary bed and BBB. In this study, it will be very important to classify analyzed specimens precisely, starting with their genetic mutation, the existence of capillary AVM, etc.
  2. Similarly, it will be very important to classify type of CCMs (sporadic, inherited form type, genetic mutations etc.). CCMs are capillary bead malformations. However, depending on the type ( inherited vs sporadic or gene mutation), the expression of tight junction proteins may vary, depending on the size and lesion progression. His should be taken into consideration and discussed.
  3. The authors should also consider that claudin-5 and occludin are expressed in arterial endothelial cells. However, the tight junction complex formed at arterial endothelial cells does not provide tightness seen at the BBB level. Thus, the authors' statement that BBB is damaged at AVM is possible, most likely in capillary AVM or as a secondary injury, but will not drive lesion progression and rupture. Thus, the link between BBB and AVM is possible but not critical for the AVM lesion. The authors should better classify the phenotype endothelial cells, whether they belong to arterial or capillaries. As this type of analysis could be very difficult due to the dedifferentiation of cells in AVM lesions, my suggestion will be to refocus the discussion from the BBB role to the potential non-canonical role of claudin-5 and occludin in the function/dysfunction of brain endothelial cells in AVM lesions.
  4. Minor concerns: A spell check is recommended. In several places in the manuscript, occludin is spelled as occluding.

Author Response

We appreciate the time and effort the reviewers have devoted to assessing our manuscript. The constructive feedback provided has been instrumental in refining our study and strengthening our conclusions

Pedrosa and coauthors investigated the tight junction profile expression in brain arteriovenous malformation (AVM) and cerebral cavernous malformation ( CCM). The analysis included human specimens of surgically resected CCMs and  AVM lesions and perilesion areas. Their results showed that bAVM had decreased Tj protein, claudin-5, and occludin expression in ruptured bAVM compared to unruptured bAVM or CCMs.           

This is a good study on an important topic.  The manuscript is well prepared, and the experiments were well designed and mapped out.  The data is convincing, and well-presented.

There are several points which are very important to address:

Comment 1: In AVM, the capillary bed, a site of the blood-brain barrier (BBB), is bypassed. However, a specific group of AVMs, the capillary AVM, could affect the capillary bed and BBB. In this study, it will be very important to classify analyzed specimens precisely, starting with their genetic mutation, the existence of capillary AVM, etc.

Response 1: Thank you for your observation. We would like to clarify that arteriovenous malformations (AVMs) are characterized by direct connections between arteries and veins, bypassing the capillary bed entirely. During surgical intervention, the neurosurgeon identifies and provides specimens, distinguishing between AVM tissue and perilesional tissue, which may contain capillaries. We have added the follow information in the methods section:

“During the surgical procedure, the neurosurgeon carefully extracted tissue sam-ples from the patient and identified their specific anatomical regions. This process in-volved distinguishing the arteriovenous malformation (AVM), characterized by direct connections between arteries and veins without the presence of capillaries, from the perilesional or perinidal tissue. The identification ensured precise categorization of the samples for subsequent analysis.”

Regarding our analysis, the tight junctions studied in the AVM samples are from the arteries. In the case of perilesional tissue, due to the nature of our sample processing, it is challenging to differentiate between arteries and capillaries. Therefore, we have generalized this classification as 'blood vessels.' To address your concern, we will specify in the manuscript that the tight junctions analyzed in AVM samples are derived from arteries. Additionally, it is important to note that all AVMs in our study are spontaneous, as none of our patients presented familial AVMs neither genetic mutation.

Comment 2: Similarly, it will be very important to classify type of CCMs (sporadic, inherited form type, genetic mutations etc.). CCMs are capillary bead malformations. However, depending on the type ( inherited vs sporadic or gene mutation), the expression of tight junction proteins may vary, depending on the size and lesion progression. His should be taken into consideration and discussed.

Response 2: We sincerely thank the reviewer for raising this important point. The reviewer is absolutely correct that it is essential to specify the type of CCMs, as this may influence the expression of tight junction proteins, among other factors. We would like to clarify that all the CCMs analyzed in our study are sporadic cases. This homogeneity in our sample reduces variability that might be introduced by inherited forms or differing genetic mutations. Consequently, our analysis focuses specifically on sporadic CCMs, providing a consistent framework for examining the role of tight junction protein expression in lesion development and progression. We will ensure that this clarification is now included in the manuscript and acknowledge that further experiments may be required to investigate variations in tight junction expression across different CCM types in future studies.

“Viable samples were obtained from sporadic 17 bAVMs and 9 CCMs.”

Comment 3: The authors should also consider that claudin-5 and occludin are expressed in arterial endothelial cells. However, the tight junction complex formed at arterial endothelial cells does not provide tightness seen at the BBB level. Thus, the authors' statement that BBB is damaged at AVM is possible, most likely in capillary AVM or as a secondary injury, but will not drive lesion progression and rupture. Thus, the link between BBB and AVM is possible but not critical for the AVM lesion. The authors should better classify the phenotype endothelial cells, whether they belong to arterial or capillaries. As this type of analysis could be very difficult due to the dedifferentiation of cells in AVM lesions, my suggestion will be to refocus the discussion from the BBB role to the potential non-canonical role of claudin-5 and occludin in the function/dysfunction of brain endothelial cells in AVM lesions.

Response 3: Thank you for your thoughtful and insightful feedback. We fully agree with the reviewer’s observations regarding the differences in tight junction complexity in arterial endothelial cells compared to the BBB level. While we acknowledge the challenges of accurately classifying endothelial cells due to dedifferentiation in AVM lesions, we would like to point out that multiple studies, including our own findings, suggest that tight junctions play an important role in the disruption of the BBB.

Although our study did not directly investigate the relationship between tight junctions and AVM rupture, we observed reduced tight junction protein expression in ruptured AVM samples. This finding highlights a potential contribution of tight junction disruption to lesion dynamics, though further studies are indeed required to elucidate the direct implications of tight junctions in AVM rupture. We will refocus the discussion in the manuscript to integrate these points, emphasizing both the potential role of tight junction proteins and acknowledging the limitations of our analysis with regard to the BBB and AVM progression.

Claudin-5 and occludin are vital tight junction proteins within endothelial cells, primarily responsible for maintaining vascular integrity and regulating the permeability of blood vessels13. While their well-known function involves forming tight junctions that control paracellular transport, emerging evidence suggests that these proteins may play additional roles in brain endothelial cells.30 These non-canonical functions offer important insights into endothelial cell dynamics and their might contribut to AVM pathology.

One notable role of claudin-5 and occludin is their involvement in signal transduction, where they may mediate cellular responses to mechanical stress and inflammatory31, that is frequently present in AVM lesions. Additionally, claudin-5 and occludin may influence endothelial cell plasticity, aiding either in maintaining cellular identity or facilitating the dedifferentiation seen in AVM lesions32. This cellular flexibility could further impact the development and progression of these malformations.

Beyond these functions, claudin-5 and occludin are also implicated in cytoskeletal interactions, contributing to endothelial cell stability and shape33. Therefore, disruptions in these interactions in AVM lesions may compromise vascular integrity and exacerbate the pathological progression of the lesions. Moreover, these proteins could play roles in abnormal angiogenesis and modulate endothelial responses in hypoxic or inflammatory environments, thereby affecting permeability and immune cell recruitment. Understanding the dysfunction and non-canonical roles of claudin-5 and occludin provides valuable perspectives for exploring therapeutic strategies targeting endothelial dysfunction in AVM pathology.

Comment 4: Minor concerns: A spell check is recommended. In several places in the manuscript, occludin is spelled as occluding.

Response 4: Thank you for bringing this to our attention. We had not noticed the error, and we have now corrected it accordingly.

Round 2

Reviewer 1 Report

Comments and Suggestions for Authors

The authors adequately revised their manuscript and accepted all the suggested changes. The manuscript was upgraded and is now acceptable for publication. The reviewer would like to thank the authors for the quality of their revision.